# Critical Size of Secondary Nuclei Determined via Nucleation Theorem Reveals Selective Nucleation in Three-Component Co-Crystals

**DOI:** 10.3390/e21111032

**Published:** 2019-10-24

**Authors:** Yang Gao, Baohua Guo, Jun Xu

**Affiliations:** Department of Chemical Engineering, Advanced Materials Laboratory of Ministry of Education (MOE), Tsinghua University, Beijing 100084, China; gaoyang910731@126.com (Y.G.); bhguo@tsinghua.edu.cn (B.G.)

**Keywords:** critical size of nuclei, secondary nuclei, co-crystal, inclusion compound, selective nucleation

## Abstract

The critical size of the secondary nuclei plays an important role in determining the crystal growth rate. In the past, the Nucleation Theorem has been applied to determine the number of molecules in the critical nuclei of a single-component crystal via variation of the crystal growth rate with dilution by the non-crystallizable component. In this work, we extend the method to the three-component co-crystal poly (ethylene oxide)/urea/thiourea inclusion compound. The theoretical crystal growth kinetics were deduced and the dependence of the radial growth rate of the inclusion compound spherulites on the mass fraction of urea in urea/thiourea was measured. The results reveal that the secondary nuclei of the poly (ethylene oxide)/urea/thiourea inclusion compound consist mainly of ethylene oxide repeating units and urea molecules. We propose that only urea molecules and ethylene oxide repeating units are selected to form the secondary nuclei while co-crystallization of the three components happens at the lateral spreading stage. As a result, the composition of the critical secondary nuclei is different from that of the bulk inclusion compound crystals. The work is expected to deepen our understanding of the nucleation of multi-component co-crystals.

## 1. Introduction

Nucleation is a central problem in the research field of crystallization [1,2,3]. According to whether the newly formed crystal nucleus is from melt or on the pre-existing surface of the same substance, nucleation can be classified as primary nucleation or secondary nucleation [4]. The latter usually occurs during the crystal growth process and determines the growth rate. If the crystal growth process adopts the surface nucleation mechanism, formation of the critical secondary nuclei with the highest Gibbs free energy acts as the rate-determining step during the crystal growth process [5,6,7,8,9,10,11].

The Nucleation Theorem based on the classical nucleation theory (CNT), proposed by Kashchiev in 1982, has been applied to determine the critical size of nuclei. Nucleation Theorem states that the derivative of nucleation work, with respect to the difference of the bulk Gibbs free energy of a crystallizable unit, equals the critical nucleus size with a negligible bias for the excess Gibbs free energy [12,13,14]. Its statistical thermodynamic foundation was proposed by Ford later [15,16]. The difference of the bulk Gibbs free energy of a crystallizable unit during the crystallization can be changed by different ratios of dilution with non-crystallizable molecules, which leads to the change of the entropy barrier of nucleation.

Following the above idea, we have adopted the Nucleation Theorem to measure the critical secondary nucleus size in a two-component co-crystal, poly (ethylene oxide) (PEO)/urea inclusion compound (IC) during its crystal growth process [17]. The PEO/urea ICs consist of guest PEO chains residing in the cavities formed by the host urea molecules [18,19,20,21,22,23,24,25,26]. By replacing a small amount of urea with N,N′-dimethyl urea with the PEO concentration unchanged, we could change the rate of secondary nucleation and determine the critical size of the secondary nuclei of the PEO/urea IC β crystal. Here, dimethylurea acted as a dilute agent, which was completely precluded from the crystal lattice. In the PEO/urea IC β crystal, the fixed stoichiometric ratio of 3/2 exists between ethylene oxide (EO) repeating units and urea molecules [22,23,24,27]; the crystallization unit is an entity containing 1 urea and 1.5 EO repeating units. Taking an entity as a unit of crystallization, the inclusion compound can be simplified as one-component crystal. A power-law relationship between the radial growth rate *G* and the mass fraction *x* of urea in the total mass of urea and the diluent was observed, and the exponent gave the critical size of the secondary nucleus. Our results revealed that a critical secondary nucleus of the PEO/urea IC β crystal consisted of 4 to 9 entities in the studied temperature range [17].

In this work, we attempt to extend the Nucleation Theorem to determine the critical size of the secondary nuclei in a three-component co-crystal system poly (ethylene oxide)/urea/thiourea inclusion compound. Thiourea is an analogue molecule of urea, and as a result, the three components co-crystallize. PEO/urea/thiourea IC prepared by electrospinning was reported by Ye et al. [28], but the crystal growth kinetics were not examined. Here we find that the spherulites of PEO/urea/thiourea IC can form by isothermal crystallization from the homogeneous PEO/urea/thiourea mixed melt, which makes the investigation of crystal growth kinetics possible.

This article is organized as follows. First, based on the Nucleation Theorem, we deduce the theoretical variation of the crystal growth kinetics of two-component co-crystal with the changed ratio of the two components. PEO/urea/thiourea IC with varied ratios of urea to thiourea molecules and fixed concentration of PEO can be considered as two-component co-crystals, so the PEO/urea/thiourea IC system can be treated with the deductions. Second, we report the crystal structure and the radial growth rates (*G*) of PEO/urea/thiourea IC spherulites with different content of urea to check the theory, and we obtain the critical nuclei size. To our surprise, the critical size of the secondary nucleus during crystal growth of the PEO/urea/thiourea IC is very similar to that of the PEO/urea IC. Finally, we will discuss the results of the crystal structure and the critical size of secondary nuclei. We propose that the critical secondary nuclei of the three-component crystal system consist of only two components, ethylene oxide segment and urea molecules, while all the three components are incorporated into the crystal lattices during the following crystal growth. Namely, the segregation of small molecules occurs only during the nucleation stage.

## 2. Theory

Considering a two-component co-crystal system consisting of crystallization units A and B, we try to deduce the crystal growth rate *G* varying with the composition. When the crystallization temperature is above the glass transition temperature and high enough, the secondary nucleation is the rate-determining step during the crystal growth. From the classical nucleation theory [5,6,7,29,30,31,32], the crystal growth rate *G* is determined by the following equation,
(1)$$G=G_{0}\exp(-W*/k_{B}T_{c})$$

In Equation (1), *G*_0_ is the pre-exponential factor, *k*_B_ is the Boltzmann constant, *T*_c_ is the isothermal crystallization temperature, and *W** is the formation work of the critical secondary nuclei.

We consider the critical secondary nuclei to be composed of components A and B. Δ*g*_1_ and Δ*g*_2_ are defined as the absolute value of the bulk Gibbs free energy difference caused by the crystallization of one A and B crystallization unit, respectively. The number of A and B crystallization unit in the critical secondary nucleus is termed as *n*_1_ and *n*_2_. The Gibbs free energy difference *W*(*n*_1_, Δ*g*_1_, *n*_2_, Δ*g*_2_) during the formation of one cluster containing *n*_1_ A units and *n*_2_ B units is expressed as follows,
(2)$$W(n_{1},\Delta g_{1},n_{2},\Delta g_{2})=-n_{1}\Delta g_{1}-n_{2}\Delta g_{2}+F_{s}(n_{1},\Delta g_{1},n_{2},\Delta g_{2})$$
where *F_s_* represents the excess Gibbs free energy during the cluster formation. If the value of Δ*g*_1_ or Δ*g*_2_ is invariable with cluster size, *W* will be a binary function of *n*_1_ and *n*_2_, and the work to form a critical secondary nucleus is equal to the maximum point *W*(*n*_1_*, Δ*g*_1_, *n*_2_*, Δ*g*_2_) of this binary function (we rewrite it as *W**(Δ*g*_1_, Δ*g*_2_)), where *n*_1_* and *n*_2_* represents the number of the crystallization units A and B in the critical secondary nucleus. So, the following equation establishes,
(3)$${\left.\frac{\partial W(n_{1},\Delta g_{1},n_{2},\Delta g_{2})}{\partial n_{i}}\right|}_{n_{1}=n_{1}*(\Delta g_{1},\Delta g_{2}),n_{2}=n_{2}*(\Delta g_{1},\Delta g_{2})}=0,i=1,2.$$

Therefore, combining the Equations (2) and (3), we have,
(4)$$\frac{\partial W*}{\partial \Delta g_{i}}={\left.\frac{\partial W}{\partial \Delta g_{i}}\right|}_{n_{1}=n_{1}*,n_{2}=n_{2}*}=-n_{i}*+{\left.\frac{\partial F_{s}}{\partial \Delta g_{i}}\right|}_{n_{1}=n_{1}*,n_{2}=n_{2}*}\approx -n_{i}*,i=1,2.$$

The capillarity approximation is adopted, namely, the interfacial free energy (per area) does not vary with the change of Δ*g* and the cluster size. This assumption in the classical nucleation theories has been generally accepted. Thus, the independence of *F_s_* on Δ*g_i_* leads to the first Nucleation Theorem described in Equation (4). The details of the deduction can be found in Kashchiev’s papers [12,13,14] and in the comments by Schmelzer [2]. Schmelzer has deduced more detailed deductions of the Nucleation Theorem and it generalized formulation, and in addition, the valid range and limits of the Nucleation Theorem are given in the comments [33]. The extension of the Nucleation Theorem to the multi-component system has also been derived by Schmelzer. In fact, Equation (4) can be derived from the Equation (27) in Schmelzer’s paper [33], assuming that the densities of the two components are the same.

The next step is to find the quantitative relation between Δ*g_i_* and the composition in the system. We define *x’* as the volume fraction of component A in the critical secondary nucleus consisting of *n*_1_* A molecules and *n*_2_* B molecules, and *x’* is approximately equal to the molar fraction *n*_1_*/(*n*_1_* + *n*_2_*). Considering that the mixing of A and B in the melt is homogeneous at the molecular level, and A molecules and B molecules randomly distribute in the critical secondary nucleus, which brings the entropy of mixing to the critical secondary nucleus, and assuming that the enthalpy of mixing A and B is zero in both the melt and co-crystal nucleus (due to the similarity between the urea and thiourea molecules when they form co-crystals in this work), the equivalent thermodynamic process is shown in Scheme 1 to calculate the absolute value of Δ*g*_1_, the Gibbs free energy change of one A molecule (or entity). Equation (5) is deduced from the equation Δ*G*_1_
*+* Δ*G*_2_ = Δ*G*_3_
*+* Δ*G*_4_ in Scheme 1, and a similar analysis can be used to deduce Equation (6).
(5)$$\Delta g_{1}=\Delta g_{10}+k_{B}T_{c}\ln(x/x^{\prime })$$
(6)$$\Delta g_{2}=\Delta g_{20}+k_{B}T_{c}\ln[(1-x)/(1-x^{\prime })].$$

In the above equations, Δ*g*_10_ and Δ*g*_20_ are the absolute values of the bulk Gibbs free energy difference caused by the crystallization of one A and B unit from the pure A and B melts, respectively. *x* represents the volume fraction of the major component A in the mixed melt of A and B. Equations (5) and (6) can be applied when *x’* is neither equal to 0 nor equal to 1; otherwise, the equations are simplified to those for the one-component crystal system.

Combining Equations (1) and (4)–(6), and *n*_1_*/*x’* = *n*_2_*/(1 − *x’*), we can calculate the fitting slope (termed as *K*) of ln*G* versus ln*x* as follows,
(7)$$\begin{array}{lll} K & = & \frac{\partial \ln G}{\partial \ln x}=\frac{\partial \ln G_{0}}{\partial \ln x}-\frac{1}{k_{B}T_{c}}\frac{\partial W*}{\partial \ln x}\approx -\frac{1}{k_{B}T_{c}}\frac{\partial W*}{\partial \ln x}=-\frac{1}{k_{B}T_{c}}(\frac{\partial W*}{\partial \Delta g_{1}}\frac{\partial \Delta g_{1}}{\partial \ln x}+\frac{\partial W*}{\partial \Delta g_{2}}\frac{\partial \Delta g_{2}}{\partial \ln x})\\ & = & n_{1}*(\frac{\partial \ln x}{\partial \ln x}-\frac{\partial \ln x^{\prime }}{\partial \ln x})+n_{2}*(\frac{\partial \ln(1-x)}{\partial \ln x}-\frac{\partial \ln(1-x^{\prime })}{\partial \ln x})\\ & = & n_{1}*-\frac{n_{1}*}{x^{\prime }}\frac{\partial x^{\prime }}{\partial \ln x}-\frac{x}{1-x}n_{2}*+\frac{n_{2}*}{1-x^{\prime }}\frac{\partial x^{\prime }}{\partial \ln x}\\ & = & n_{1}*-\frac{x}{1-x}n_{2}* \end{array}$$

Equation (7) gives the relationship between ln*G* versus ln*x* expressed by the parameters of the critical secondary nucleus. The prefactor *G*_0_ may vary with the composition *x*. For the two-component system, we can first consider the following two extreme cases: if only one component (with the volume concentration noted as *x*) is involved in the secondary nucleation, *G*_0_ is proportional to *x*. If the two components are simultaneously involved in nucleation and the nucleation rates of the two components are the same, *G*_0_ does not change with *x*. The other cases lie between the above two extreme cases, so the partial derivative of ln*G*_0_ with respect to ln*x* ranges from 0 to 1. Thus, considering the possible variation of *G*_0_ with *x*, the difference between the slope *K* and the number of molecules (or entities) in the critical nuclei is less than 1, and we can omit this term. When *n*_2_* equals zero, Equation (7) is simplified to the case of the nucleation of a single-component crystal.

To apply Equation (7) to the three-component PEO/urea/thiourea IC system, we regard an entity consisting of 1 EO with 1.5 urea and that of 1 EO with 1.5 thiourea as the two types of crystallization units A and B, respectively. The stoichiometric ratio of EO units to small molecules in the PEO/urea inclusion compound is 3/2, and each unit cell averagely contains 1.5 EO and 1 urea molecule [22,23,24,27]. We use 1 small molecule and 1.5 ethylene oxide as an entity of crystal nucleation. Consequently, the three-component co-crystal PEO/urea/thiourea IC can be treated as a two-component A/B co-crystal. For simplicity, we approximate the volume ratio by mass ratio since urea and thiourea have similar densities. So, in the later section, the mass ratio of the PEO/small molecules is fixed to 1/0.91, which is the same as the EO/urea mass ratio in PEO/urea IC β crystal, and *x* represent the mass fraction of urea in the total mass of urea/thiourea.

## 3. Experimental Section

### 3.1. Sample Preparation

Poly (ethylene oxide) (PEO) with a viscosity-average molecular weight (*Mv*) of 100,000 g/moL and urea were both purchased from the Sigma-Aldrich Company. Thiourea was purchased from the Aladdin Company. All the materials were used without further purification.

The mass ratio of EO to the sum of urea and thiourea was fixed at 1/0.91, which is equal to the mass ratio of EO to urea in the PEO/urea IC β phase. The mass ratio of urea to thiourea was set from 10/0 to 0/10. Complete dissolution of PEO (0.2 g), urea, and thiourea (0.182 g of the latter two in total) in the deionized water (3 mL) was accelerated by heating and stirring. Then the water solution of PEO/urea/thiourea was rapidly cooled down and crystallized in liquid nitrogen to ensure the homogeneous composition in the obtained ice block. After being freeze-dried for 48 h to completely remove the water solvent, a white spongy solid block of homogenous PEO/urea/thiourea mixture was obtained.

The crystals of the PEO/urea/thiourea inclusion compounds with different compositions were prepared by isothermal crystallization at 70 °C, after being melted at 155 °C for 15 s, for the relevant characterizations.

### 3.2. Characterization

Each film of PEO/urea/thiourea IC crystal with a thickness of about 200 μm was prepared by sandwiching milligrams of the as-prepared bulk sample between a cover glass and a piece of polyimide film. The isothermally crystallized samples were obtained by quenching to 70 °C, and were held at the same temperature for enough time (usually less than 30 min) after melting at 155 °C for 15 s on a hot-stage. After cooling down to the room temperature, the cover polyimide film was carefully removed to obtain a bare, flat surface of the PEO/urea/thiourea IC crystals for one-dimensional X-ray diffraction (XRD) measurement, which was carried out on a diffractometer (Riguka SmartLab, Rigaku, Tokyo, Japan) with a scanning rate of 10°/min with 2θ step of 0.01° from 5 to 50° under the Ni-filtered Cu *K*α radiation (40 kV, 30 mA).

The thermal behaviors of the PEO/urea/thiourea IC crystals were studied by differential scanning calorimeter (Shimadzu DSC-60, Shimadzu, Kyoto, Japan) under nitrogen atmosphere. An indium standard was used for calibration before the measurement. All the samples were melted at 150 °C for 1 min, and then quenched to 70 °C and isothermally crystallized completely for 30 min, followed by heating to 160 °C with a rate of 10 °C/min to measure the melting points and enthalpies of fusion.

To observe the spherulite morphology and to measure the radial growth rates, a polarized optical microscope (POM) (Olympus BX41P, Olympus, Tokyo, Japan) equipped with a CCD digital camera (Motic Moticam Pro 282A, Motic, Xiamen, China) was used. A sample sandwiched between two glass slides of 10 mm × 10 mm was first melted at 155 °C for 15 s to eliminate the thermal history, and then pressed to a film with the thickness around 10 μm. Each growth rate of spherulites was measured for three different samples and the averaged value of growth rate and standard deviation were calculated.

## 4. Results and Discussion

### 4.1. Crystalline Structure of the Peo/Urea/Thiourea Ternary Inclusion Compounds

The wide-angle X-ray diffractograms of a series of PEO/urea/thiourea IC samples are presented in Figure 1a. The mass ratio of PEO and small molecules (urea plus thiourea) is fixed to 1/0.91, with the mass ratio of urea to thiourea varying from 10/0 to 1/9. All the diffractograms appear almost the same, except that the positions of the main diffraction peaks shift to the low-angle direction steadily with the increasing content of thiourea molecules (Figure 1b). The results reveal that the crystal structure is similar in all the samples, namely, the crystal structure of all the samples are the same as the orthorhombic system of the PEO/urea IC β phase. The main diffraction peaks are indexed in Figure 1.

From the position shift of the diffraction peaks, we can estimate the expansion rate of the spacing of the diffraction planes in PEO/urea/thiourea IC crystals. The interplanar spacings of the different (*hkl*) planes, *d_hkl_*(*x*), with *x* indicating the mass fraction of urea molecules in urea/thiourea, are calculated based on the Bragg’s law in Equation (8). The percentage of expansion *α_hkl_* is defined by dividing the slope of the linear fitting of the expansion distance *d_hkl_*(*x*) versus *x* by the interplanar spacing of PEO/urea IC crystal without thiourea molecules *d_hkl_*(*x* = 1) (Figure 1c and Equation (9)). The expansion percentages of the different crystal faces range from 3% to 7% (Figure 1d). We conclude that the continuous change of the interplanar spacings is due to the gradual addition of the thiourea molecule, which suggests that the thiourea molecules replace the urea molecules stochastically and uniformly. In other words, PEO/urea/thiourea IC is a ternary isomorphic system.
(8)$$d_{hkl}(x)=\frac{\lambda }{2\sin\theta_{hkl}(x)}$$
(9)$$\alpha_{hkl}=-\frac{1}{d_{hkl}(1)}\frac{d}{dx}d_{hkl}(x).$$

### 4.2. Thermodynamic Stability of the PEO/Urea/Thiourea Ternary Inclusion Compounds

The DSC heating curves of the ternary isomorphic PEO/urea/thiourea IC completely crystallized at 70 °C are shown in Figure 2a. With the addition of the thiourea component, the melting point *T*_m_ of the PEO/urea/thiourea IC decreased first, and then increased (Figure 2b), which reflects the same tendency of the thermodynamic stability. The enthalpy of fusion of the PEO/urea/thiourea ternary isomorphic IC decreased first, and then it leveled off to about 100 J/g (Figure 2b). The obvious decrease of the enthalpy of fusion in the sample with the urea mass fraction in urea/thiourea of 0.1 is mainly caused by the separate crystallization of some thiourea molecules.

We can conclude that the TU molecules participate in the crystallization of IC. If the TU molecules did not participate in the crystallization of IC, the theoretical enthalpy of fusion of the three-component samples should be *x* times of that of PEO/U IC (141.4 J/g), which is much lower than the values in Figure 2b obtained from the DSC thermograms. These support that the PEO, urea, and thiourea have formed the ternary isomorphic inclusion compound system.

In a strict isomorphic system, the melting point and enthalpy of fusion should change linearly with crystal composition [34,35,36], but our PEO/urea/thiourea IC system does not follow this law. There are two possible reasons: (1) The two types of host small molecules do not distribute homogeneously at the molecular level in the IC crystals, but exhibit the homogeneous mixing for the XRD characterization; (2) when urea and thiourea molecules crystallize together, some defects in the IC crystals might form. The density of defects would be the largest at the mediate ratio of urea/thiourea, so the melting point of IC shows a V-shape tendency with the content of urea in urea/thiourea and the enthalpy of fusion decreases with the increasing of thiourea content.

### 4.3. Spherulite Morphologies and Radial Growth Rate of PEO/Urea/Thiourea IC

Here we focus on the crystal growth kinetics of PEO/urea/thiourea IC so as to explore the effect of the urea/thiourea composition on the radial growth rate of PEO/urea/thiourea IC spherulites.

Figure 3 presents the POM images of the spherulites of PEO/urea/thiourea IC with different compositions isothermally crystallized at 70 °C. No phase separation in the mixed melt suggests that PEO chains, urea, and thiourea molecules form a mixed melt at the molecular level, which has also been proved by Ye et al. via infrared spectroscopy [28]. The hydrogen-bond network between the three components prevents the occurrence of phase separation in the mixed melt. Therefore, we believe that the concentration of urea molecules at the crystal growth front of spherulite is the same as the initial global concentration.

Figure 4 plots the measured radial growth rate of PEO/urea/thiourea IC spherulites varying with the content of urea at different isothermal crystallization temperatures. The urea mass fractions in the urea/thiourea host small molecules are from 100% to 60%. The results of samples with the urea mass fractions in urea/thiourea lower than 60% are not shown here because of the large uncertainties of the data. The radial growth rate decreases with the increasing temperature in each sample, and its special tendency supports the surface nucleation mechanism for spherulite growth [4,17].

Figure 5 illustrates the relationship between the spherulite radial growth rates *G* of IC and the mass fractions *x* of urea in urea/thiourea. Figure 6 shows the plot of ln*G* and ln*x*. Deduced from the theory based on the Nucleation Theorem, ln*G* is proportional to ln*x* of the crystallizable component in the binary blend system if the other component does not participate in the crystallization, as confirmed by our previous work on the PEO/urea/dimethylurea blend system [17], and Equation (7) dominates the crystallization kinetics of the double-component crystallization system. However, to our surprise, the preferable linear dependence of ln*G* on ln*x* still occurs in the PEO/urea/thiourea ternary isomorphic IC system. The linear fitting of ln*G* versus ln*x* gives the slopes ranging from 5 to 12 at isothermal crystallization temperatures from 60 °C to 85 °C with the square of linear correlation coefficients all above 0.97, which inspires us to consider the reason behind the crystal growth kinetics and the detailed secondary nucleation mechanism.

### 4.4. Critical Size of the Secondary Nucleus of PEO/Urea/Thiourea IC

According to Equation (7), if the number ratio of urea/thiourea in the critical secondary nucleus is equal to the volume ratio (approximately equal to the mass ratio) of urea/thiourea in the three-component melt before crystallization, i.e., *n*_1_*/*n*_2_* ≈ *x*/(1 − *x*), the value of *K* should be zero. However, our experimental results demonstrate that the fitting slope *K* is a non-zero value, which does not change with *x* when *x* ranges from 60% to 100%. If *n*_2_* adopts a non-zero value, *K* will vary with *x* obviously according to Equation (7). Consequently, according to our experimental results that ln*G* shows linear dependence on ln*x*, i.e., *K* is almost constant for all *x* at each crystallization temperature, the simplest explanation is that *n*_2_* is zero (namely, *x*’ = 1) and the slope *K* equals *n*_1_* for all *x*. Specifically, the critical secondary nucleus does not contain thiourea molecules. We propose that during the crystal growth process of PEO/urea/thiourea IC, the critical secondary nuclei consist mainly of urea molecules and EO units. Namely, the nucleation event selects urea and EO units from the three-component PEO/urea/thiourea melt, implying a stronger nucleating ability of urea with EO units compared to thiourea molecules. The reasonability of our proposition will be discussed in the following text.

According to our previous research results, the critical secondary nucleus of PEO/urea IC consists of 3 to 9 urea molecules when the crystallization temperature ranges from 65 to 85 °C [17]. The number of urea molecules in the critical secondary nucleus of the three-component PEO/urea/thiourea IC is similar to that of the two-component IC of PEO/urea, confirming that only the urea molecules are selected to form the critical secondary nuclei with EO repeating units. In other words, the nucleation ability of urea molecules to form an inclusion compound with ethylene oxide units is stronger than thiourea molecules, so the selection of urea molecules occurs at the secondary nucleation stage. At the later lateral spreading growth stage used to complete the growth of each layer on the growth front, EO, urea, and thiourea can co-crystallize together. The diagrammatic sketch of the above microscopic process is drawn in Figure 7. Because the secondary nuclei only take a tiny percentage in the samples, X-ray diffraction could not detect the existence of the segregated PEO/urea IC nuclei. A larger size in the critical secondary nucleus compared with the PEO/urea system is probably caused by the more difficult diffusion of urea molecules in the melt in the presence of thiourea.

Another piece of evidence supporting the stronger nucleating ability of urea with EO units comes from the fact that introducing a small amount of urea molecules (e.g., 10%) can make the crystal structure of PEO/urea/thiourea IC adopt the orthorhombic crystal system of the PEO/urea IC β phase, as shown in the above Figure 1a, rather than the monoclinic crystal system of the PEO/thiourea IC α phase [28,37,38].

## 5. Conclusions

Via theoretical deductions from the Nucleation Theorem based on the entropy effect due to dilution, we establish a method to determine the critical size of secondary nuclei in an isomorphic co-crystal system (the composition of each component in the crystal is the same as that in the melt) from the variation of the radial growth rates of spherulites in the composition. The method is further applied to the three-component system of inclusion compounds formed between small molecules and polymer chains. In the work presented here, we adopt a method to determine the critical size of the secondary nuclei of three-component co-crystal PEO/urea/thiourea inclusion compounds. After the gradual replacement of urea by thiourea molecules with fixed PEO concentration, the crystal structure of PEO/urea/thiourea IC maintains the orthorhombic system of PEO/urea IC β crystals, though the cell lattice parameters expand a bit, with less than 3% on average. This result indicates that the co-crystal of the three components adopts an isomorphic crystal structure. The thermodynamic stability of PEO/urea/thiourea IC and the radial growth rate of spherulites first decrease and then increase with the increasing content of thiourea molecules. Furthermore, the addition of a small amount of urea (such as 10%) can change the crystal structure of the three-component IC from α form of PEO/thiourea IC to the crystal structure close to that of PEO/urea β IC.

A good power-law relationship is observed between the radial growth rate *G* of the PEO/urea/thiourea IC spherulites and the mass fraction *x* of urea in urea/thiourea when the isothermal crystallization temperature *T*_c_ ranges from 60 to 85 °C. The slope of the linear fitting curve increases from 5 to 12 with increasing temperature, but does not vary with the urea mass fraction. The results reveal that during the crystal growth process of PEO/urea/thiourea IC, the critical secondary nuclei consist mainly of urea molecules and EO units, but in the lateral spreading to complete the growth of each layer, the three components can co-crystallize together to form the three-component inclusion compound. Our finding that the selection of a component occurs at the secondary nucleation stage rather than the following lateral spreading growth of the co-crystal system deserves further study.

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
