# Peer review of "Critical Size of Secondary Nuclei Determined via Nucleation Theorem Reveals Selective Nucleation in Three-Component Co-Crystals"

_entropy, 2019, doi:10.3390/e21111032_

Round 1

Reviewer 1 Report

This paper describes a rigorous theoretical treatment for the nucleation in the crystallization process. Based on this theory the authors determined the size of secondary nuclei during the crystal growth of PEO/urea/thiourea system. The experiments were carefully designed and the results seem quite reasonable in light of their own theory. This is a sound paper and I would recommend it for publication in the present form.

Author Response

Response to the reviewer 1:

This paper describes a rigorous theoretical treatment for the nucleation in the crystallization process. Based on this theory the authors determined the size of secondary nuclei during the crystal growth of PEO/urea/thiourea system. The experiments were carefully designed and the results seem quite reasonable in light of their own theory. This is a sound paper and I would recommend it for publication in the present form.

Answer: Thank you for the review process and appreciation of our work!

Reviewer 2 Report

The authors discussed the secondary nucleation of the co-crystals of poly(ethylene oxide)/urea/thiourea. Although the crystallization behavior of this ternary system may not be investigated so far, the framework of theoretical background contains unreliable discussion, which makes the conclusions uncertain.

The excess free energy Fs contains the contributions of the surface free energy of crystals, which dominates the size of the secondary nuclei discussed around line 274. The framework in this manuscript cannot be based on the discussion on the sizes. The value of n* is also a function of surface free energy and composition. The presented framework ignored the contributions of the composition to K through n*.

In discussing K in eq. 7, the dependence of the prefactor G0 on the composition was not taken into account. The simplest model would be that G0 is proportional to the volume fraction of a component. The two-entity model of this paper may require different G0 for the different entities. This can be a possible reason for the discrepancy of the slope of Fig. 6.

The entities containing a fractional number of molecules are difficult to be accepted. The relation between the unit cell and the entities was not explained.

The entropy effect described in lines 201-202 assumes that the entropy of crystals is unaffected by adding the third component. However, this is not applicable to co-crystals or solid solutions. They contain the contributions of entropy of mixing.

Polarized-light microscopy is an unsuitable method to show homogeneity of a melt (line 218).

The following lines were not explained sufficiently.

(a) Reason 1 in lines 208-210.

(b) The descriptions in lines 266-267.

Round 2

Reviewer 2 Report

The explanations about the zero entropy of mixing for co-crystals were not found in the manuscript, which should contain similar descriptions as in the authors’ response.

The authors described that “the difference between the slope K and the critical nuclei size is less than 1” on lines 134–135. The slope K and the nuclei size are different physical quantities. They cannot be compared with each other and the “difference” cannot be obtained.

The authors should comment on the following points in the discussion part of the manuscript. Based on the scheme about the entropy of mixing for co-crystals indicated in the authors’ response, different values in x and x’ result in non-zero entropy of mixing. If the secondary nuclei consist of PEO and only urea, x’ = 1 for all x upon secondary nucleation, which results in x’ ≠ x.
